# A Nutritional Supplement as Adjuvant of Gabapentinoids for Adults with Neuropathic Pain following Spinal Cord Injury and Stroke: Preliminary Results

**DOI:** 10.3390/healthcare11182563

**Published:** 2023-09-16

**Authors:** William Raffaeli, Giorgio Felzani, Michael Tenti, Luca Greco, Maria Pia D’Eramo, Stefania Proietti, Giovanni Morone

**Affiliations:** 1ISAL Foundation, Institute for Research on Pain, 47921 Rimini, Italy; william.raffaeli@fondazioneisal.it; 2San Raffaele Institute of Sulmona, 67039 Sulmona, Italy; 3Unit of Clinical and Molecular Epidemiology, San Raffaele University, 00166 Rome, Italy; 4Department of Life, Health and Environmental Sciences, University of L’Aquila, 67100 L’Aquila, Italy

**Keywords:** neuropathic pain, spinal cord injury, stroke, supplements, gabapentinoids, pregabalin, adjuvant

## Abstract

Gabapentinoids are first choice drugs for central neuropathic pain (CNP) despite limited evidence of efficacy and side effects affecting therapy outcomes. Nutraceuticals could improve their efficacy and tolerability. Our aim is to investigate the effect of NACVAN^®^, in addition to gabapentinoids, on pain symptomatology in CNP patients. The effect of 6 weeks of treatment of NACVAN^®^ was preliminary observed among 29 adult inpatients with spinal cord injury (SCI) or stroke-related CNP recruited to the experimental group. Pain intensity, neuropathic pain, and quality-of-life were measured at baseline (T0) and after 3 (T1) and 6 weeks (T2). Change in each outcome over time was assessed through a repeated measures analysis of variance or Wilcoxon matched-pairs test. Preliminary results show a significant reduction in pain intensity (T0 → T1, *p* = 0.021; T0 → T2, *p* = 0.011; T1 → T2, *p* = 0.46), neuropathic symptoms (T0 → T1, *p* = 0.024; T0 → T2, *p* = 0.003), and evoked pain (T0 → T2, *p* = 0.048). There were no significant reductions in other neuropathic pain dimensions and in quality-of-life components. No side-effects were detected. NACVAN^®^ could have a beneficial adjuvant effect when used as an add-on to gabapentinoids in patients suffering from CNP due to SCI or stroke, with no adverse effect. Future analysis on a larger sample, compared with a placebo condition, could confirm these preliminary results.

## 1. Introduction

The Neuropathic Pain Special Interest Group (NeuPSIG) of the International Association for the Study of Pain (IASP) defines neuropathic pain (NP) as a type of pain caused by an injury or pathology of the somatosensory system, either peripherally or centrally [1]. It affects 6–8% of the general population, with a prevalence similar to that of other diseases such as diabetes and asthma. It can underlie different syndromes with different pathophysiologies, such as diabetic neuropathy, post-herpetic neuralgia, or pain from myelic or stroke injury outcomes [2,3]. NP can be described as spontaneous or evoked, as a burning or cold sensation or with electric-shock-like qualities, or as a paresthesia or a sense of numbness. Negative symptoms such as hypoalgesia, hypopallesthesia, or tactile hypoesthesia may also be present [4]. NP is often severe and difficult to manage adequately; therefore, it negatively affects the functioning of patients and their quality of life [5].

### 1.1. Neuropathic Pain from Stroke Outcomes or Spinal Cord Injury

When NP develops because of an injury or a pathology affecting the brain, brainstem, or spinal cord, it is defined central neuropathic pain (CNP) [1]. It is often a consequence of stroke or spinal cord injury (SCI). Data from population-based studies and recent meta-analyses suggest that CNP occurs in 11% of stroke patients [2] (even in more than 50% of medullary or thalamic stroke cases) [6] and in 53% of patients with SCI [7]. In more than 50% of patients, central post-stroke pain develops within the first month after the event, in 31% of patients within a month, and within the first year in 41% of cases [6]. The onset of CNP following SCI seems to be more common after 6 months after the injury [7].

### 1.2. The Management of Neuropathic Pain from Stroke or Spinal Cord Injury Outcomes: The Role of Gabapentinoids and Pregabalin

Central neuropathic pain (CNP) following stroke or SCI is among the most complicated to manage. Pain resolution is unlikely, and clinically meaningful relief often corresponds to a two-point reduction in pain intensity on a numerical rating scale (NRS) at best. Treatment recommendations come from very few randomized controlled trials (RCTs) or algorithms created for the management of peripheral neuropathic pain, but patients with CNP are often unresponsive to pharmacological agents used to treat peripheral neuropathic pain [8].

Gabapentinoids are the drugs that have shown greatest efficacy in the treatment of NP in RCTs [9]. In particular, the gamma-aminobutyric acid (GABA) analogue pregabalin has shown the best efficacy and tolerability in this setting and it is recommended by international guidelines for several types of peripheral neuropathic pain in adults [10]. However, a recent Cochrane review found low-quality evidence regarding the efficacy of pregabalin (600 mg) in CNP following stroke or SCI [11]. Despite the limited amount of RCTs regarding pharmacotherapy of CNP following stroke or SCI and the low quality of the evidence highlighted by the review of Derry and colleagues (2019) [11], the findings of other reviews suggest that pregabalin is still one of the first-choice treatments for these conditions [6,8].

### 1.3. Difficulties in the Use of Gabapentinoids and Pregabalin in Patients with NP

Gabapentinoids and pregabalin are generally well-tolerated but can cause adverse effects including dizziness, drowsiness, peripheral edema, weight gain, asthenia, headache, and dry mouth [12]. These side effects are generally mild, well-tolerated, and reversible upon stopping the drug consumption. However, in some people these side effects may cause a dose reduction, a therapy discontinuation or termination, or a difficulty in reaching the effective dose [4]. Moreover, in CNP conditions, gabapentinoids are sometimes prescribed together with other drugs with central side effects, and such polypharmacology may affect neurological function in subtle but clinically relevant ways [8,13]. Evidence also suggests that some patients may abuse gabapentinoids; in this regard, the dose-dependence of pregabalin may be particularly problematic [14].

There is, therefore, a need for strategies to improve the efficacy and tolerability of gabapentinoids and pregabalin.

### 1.4. The Role of Nutraceuticals as Adjuvants in the Treatment of NP

Different studies have suggested the potential role of nutraceuticals as adjuvants of pharmacotherapy in the field of NP. For example, B complex has important neuroprotective and neuroinflammatory effects, and numerous studies have shown that B1 (thiamine), B6 (pyridoxine), and B12 (cobalamin) exercise an analgesic effect in different NP conditions, especially when taken in combination [15]. These vitamins specifically inhibit certain pathophysiological processes involved in NP with a dose-dependent analgesic effect: higher dosages, in fact, correspond to more immediate and sustained benefits on pain [16]. Vitamins B1, B6, and B12 have such an important role in the management of neuropathies that they are recognized as “neurotrophic B-vitamins” [17]. Of note, supplementation with B-vitamins represents a viable therapeutic option not only for neuropathies related to deficiency of these vitamins, but also in cases of nondeficient neuropathies [18]. 

Curcumin, a phenolic compound derived from Curcuma longa, has been found to have an analgesic effect in several pain conditions, including osteoarthritis, chronic post-surgical pain, and rheumatoid arthritis [15,19]. Curcumin may also have a specific effect on NP due to its affinity with the transient receptor potential vanilloid 1 (TRPV1), whose role in the genesis of NP has been found in several studies on animal models [20,21,22].

Similar results have been obtained for N-acetylcysteine (NAC), an endogenous molecule with high antioxidant properties that has shown efficacy in reducing NP by inhibiting matrix metalloproteinases [23]. NAC appear to be useful in NP conditions also because of its ability to enhance activation of Glu2/3 receptors for glutamate [24]. A recent study involving patients with diabetic neuropathy suggested that the administration of NAC as adjunctive therapy to pregabalin significantly increased pain control compared to pregabalin alone [25].

The methanolic extract of Boswellia dalzielii, a tree plant prevalent in the Maghreb region, could be an additional, useful therapeutic agent for both the prevention and therapy of different pain conditions, as evidenced by a study recently conducted in animal models [26]. The study of the therapeutic potential of Boswellia extracts has greatly increased since the 1990s, and today its role in the treatment of osteoarthritis pain is well established [15]. In addition, the use of Boswellia extracts can help in reducing the use of non-steroidal anti-inflammatory drugs (NSAIDs), contributing to more efficient control of their potential side effects [27].

The combination of compounds described so far is contained in a new commercially available nutraceutical—NACVAN^®^—consisting of NAC (300 mg), turmeric extract (211 mg), Boswellia extract (200 mg), and vitamins B1 (10 mg), B6 (5 mg), and B12 (2.5 mcg). The use of this compound, therefore, could have potential benefits in different pain conditions, especially those with neuropathic characteristics, as an adjuvant of the standard pharmacotherapy. Indeed, convincing evidence has been already provided for the use of the combined use of gabapentinoids, with some of the substances within NACVAN^®^ for some NP conditions [15].

The aim of the present study is, therefore, to investigate the effect of NACVAN^®^, administered in addition to pregabalin or other gabapentinoids, on pain intensity and different dimension of neuropathic pain in patients with CNP due to SCI or stroke.

## 2. Materials and Methods

### 2.1. Study Design

This study employs an uncontrolled design. The research project envisaged adopting a non-randomized, placebo-controlled, single-blind design with two groups: NACVAN^®^ or placebo. Unfortunately, several spinal unit (SU) inpatients and people on the waiting list for hospitalization in the SU contracted the SARS-CoV-2 pneumonia (COVID-19) during the study period. This has significantly lengthened hospitalization times, hindering the recruitment of new patients. An interim power analysis (see the “Data analysis” section) was performed after 24 subjects had been enrolled in the experimental group, revealing that the sample was adequately powered to evaluate the study outcomes in a repeated-measures, within-subject research design. According to these statistical considerations and owing to the above-mentioned difficulties in patients’ recruitment, here we present the preliminary results regarding the effect of the adjuvant use of NACVAN^®^ on adult inpatients with CNP due to SCI or stroke included in the experimental group only, without any reference to the placebo control group.

### 2.2. Participants

The present trial was conducted at the spinal unit (SU) of the institute “San Raffaele Sulmona” (Abruzzo, Italy). The study was carried out on adult inpatients suffering from CNP secondary to SCI or stroke, treated with pregabalin or other gabapentinoids according to international guidelines [28,29] but without sufficient pain relief. 

Participants were consecutively assessed and recruited by a neurologist or a physiatrist of the SU. 

Eligibility criteria included: (a) patients of both sexes > 18 years; (b) patients with CNP for at least 3 months secondary to SCI or stroke, treated with pregabalin or other gabapentinoids according to international guidelines; (c) patients unresponsive or intolerant to other analgesics and/or adjuvants indicated by international guidelines; (d) patients able to express a conscious and autonomous consent. Exclusion criteria included: (a) intolerance to pregabalin or other gabapentinoids; (b) intolerance to the NACVAN^®^ compounds; (c) patients with cancer or severe psychiatric diseases; (d) pregnancy. 

Between July 2022 and March 2023, 29 participants were considered for participation in the study. Researchers explained study aims to the participants and collected their written informed consent.

### 2.3. Intervention

One tablet of NACVAN^®^ (Ca.Di.Group S.r.l., Rome, Italy) containing NAC (300 mg), Curcuma longa extract (211 mg), Boswellia serrata extract (200 mg), and vitamins B1 (10 mg), B6 (5 mg), and B12 (2.5 mcg) were administered twice daily for 6 weeks. The NACVAN^®^ was included within the National Register for Food Supplements by the Italian Ministry of Health. The NACVAN^®^ was added to patients’ basic therapy, characterized by pregabalin or gabapentin (see Table 1).

### 2.4. Measures

Socio-demographic and clinical characteristics of participants were collected by a trained psychologist (L.G.) using an ad hoc case report form (CRF). Clinical characteristics included information about onset of pain, pain site(s), pain frequency, number of surgery interventions related to SCI, pharmacotherapy, and its perceived efficacy.

Treatment outcome measures were administered at baseline (T0), after 3 weeks of NACVAN^®^ intake (T1, half treatment) and after 6 weeks of NACVAN^®^ intake (T2, end of the treatment). All questionnaires were administered via face-to-face interview with the psychologist of the SU.

The primary outcome measures were the perceived intensity of pain and NP. The perceived intensity of pain was measured through a numerical rating scale (NRS) based on a 0–10 response format, with 0 for “no pain” and 10 for “the worst pain imaginable”. The NRS demonstrated good validity and reliability across a wide range of painful conditions, including chronic neuropathic pain [30]. The different dimensions of NP were measured through the Italian version of the Neuropathic Pain Symptom Inventory (NPSI) [31]. It is a self-administered questionnaire including 10 pain descriptors [all ranging from 0 (none) to 10 (worst imaginable)] and 2 temporal items. The NPSI returns a total intensity score (ranging from 0 to 100) and five scores corresponding to different clinically relevant dimensions of neuropathic pain, i.e., burning (superficial) spontaneous pain, pressing (deep) spontaneous pain, paroxysmal pain, evoked pain, and paraesthesia/dysesthesia. The NPSI has been found to be a valid and reliable measure for neuropathic symptoms in an Italian population with peripheral nerve diseases [31].

Secondary outcome measures included the Italian version of the short-form health survey 12 [32] and the standard 7-point patient global impression of change (PGIC). The SF-12 is a brief self-report measure of subjective health status consisting of 12 items generating physical (PCS) and mental component summary (MCS) scores that reflect physical and mental domains of health-related quality of life (HR-QOL), respectively. The PCS and MCS scores range from 0, denoting the lowest level of HR-QOL, to 100, denoting the highest level of HR-QOL. The PGIC was a test of global impressions of improvement based on a 7-point scale, by which patients considered any changes observed in their own pain condition from the beginning of the treatment with an evaluation ranging from “No change (or condition is worse)” to “A great deal better, and a considerable improvement that has made all the difference”. The PGIC was administered only at the end of the NACVAN^®^ intake period (T2).

Participants were instructed to report any adverse events to the principal investigator or nurses of the SU at any time during the NACVAN^®^ intake period. Side effects were registered within the above-mentioned CRF.

### 2.5. Data Analyses

An interim a priori power analysis performed using G*Power (v. 3.1.9.7) revealed a minimum sample size of N = 20 to detect an effect size of d = 0.3 with a power of 0.80 in a repeated-measures analysis of variance (ANOVA) (α = 0.05; three measurements; correlation among repeated measures = 0.50). The effect size of the a priori power analysis was informed by the smallest effect size detected by an earlier trial demonstrating the effects of a non-pharmacological intervention on clinical outcomes in SCI patients [7].

Participant socio-demographic and clinical characteristics, as well as PGIC, were reported using descriptive statistics [mean (SD), frequency (percentage)]. Preliminary controls for normality (Shapiro–Wilk test) were performed. Change in each outcome over time (T0–T1–T2) was assessed through a repeated measures analysis of variance (ANOVA) or Wilcoxon matched-pairs test, where appropriate. All statistical analyses were performed using SPSS v. 25.0 for Windows (IBM Corp., Armonk, NY, USA). All tests were two-tailed, and a *p*-value of less than 0.05 was considered statistically significant.

## 3. Results

During the study period, 29 patients were considered eligible based on the inclusion/exclusion criteria. Of these, 24 completed the study successfully, while 5 participants dropped out before the end of the NACVAN^®^ intake period for an early discharge from the SU.

Baseline socio-demographic and clinical characteristics of the final sample are shown in Table 1. All participants were of Caucasian ethnicity.

All 24 patients completed the questionnaires in full. Table 2 shows mean values and standard deviations, at any time point, for the outcome variables, i.e., pain intensity, neuropathic pain dimensions measured through the NPSI, and physical and mental component of HR-QOL measured through the SF-12.

The data distribution was normal except for pain intensity (NRS) at T0; the total score of the NPSI at T0, T1, and T2; the NPSI subscale “paroxysmal pain” at T0, T1, and T2; the NPSI subscale “burning (superficial) spontaneous pain” at T0, T1, and T2; and the PCS score of the SF-12 at T0.

The primary outcome measures perceived intensity of pain and NP evaluated with the NRS and the NPSI, respectively. The mean pain intensity score was 7.79 at baseline (T0), 7.25 after 3 weeks (T1), and 6.79 after 6 weeks (T2). The reduction in pain intensity was significant from T0 to T1 (Z = −2.31; *p* = 0.021), from T0 to T2 (Z = −2.54; *p* = 0.011), and from T1 to T2 (Z = −2.00; *p* = 0.46) (Figure 1). Individual patients’ pain intensity scores are presented in Figure 2.

The mean NPSI total score was 29.18 at baseline (T0), 27.25 after 3 weeks (T1), and 25.77 after 6 weeks (T2). The reduction in the NPSI total score was significant from T0 to T1 (Z = −2.26; *p* = 0.024) and from T0 to T2 (Z = −2.98; *p* = 0.003), but not significant from T1 to T2 (Z = −1.83; *p* = 0.68) (Figure 3). Individual patients’ NPSI total scores are presented in Figure 4.

The mean score of the NPSI subscale “evoked pain” was 12.38 (±7.76) at baseline (T0), 11.56 (±7.23) after 3 weeks (T1), and 10.68 (±7.15) after 6 weeks (T2). The reduction in the NPSI subscale “evoked pain” score was significant from T0 to T2 (*p* = 0.048), but not significant from T0 to T1 (*p* = 0.376) nor from T1 to T2 (*p* = 0.213) (Figure 5). Individual patients’ NPSI evoked pain subscale scores are presented in Figure 6.

There were no significant reductions in scores of the NPSI subscales “burning (superficial) spontaneous pain”, “pressing (deep) spontaneous pain”, “paroxysmal pain”, and “paraesthesia/dysesthesia”, and in physical (PCS) and mental component summary (MCS) scores of the SF-12.

Figure 7 shows the PGIC after 6 weeks of NACVAN^®^ intake. Overall, 33.3% of patients did not recognize any change after the NACVAN^®^ intake, while 66.7% of them declared to feel somewhat better after the NACVAN^®^ supplementation.

Participants reported no side-effects during the NACVAN^®^ intake period.

## 4. Discussion

The aim of the present study was to evaluate the effects of a new commercially available nutraceutical, used as an add-on to gabapentinoids, on pain intensity and different dimension of NP in patients affected by CNP due to SCI or stroke. Results suggest that NACVAN^®^ could have a beneficial adjuvant effect in these patients, with no adverse effect. Specifically, it could help in reducing the perceived intensity of pain and NP symptoms, in particular evoked pain, already after 3 weeks of intake. 

Our results confirm pre-clinical findings on the antinociceptive mechanisms of nutraceuticals and, specifically, of the NACVAN^®^ compounds. A recent systematic review and meta-analysis already highlighted how nutraceutical compounds can lead to an improvement in preclinical NP models [33]. Particularly, the authors considered the effect of fifteen natural compounds, mainly antioxidants, and showed their significant effect in reducing NP (measured through thermal hyperalgesia or mechanical allodynia/hyperalgesia) in animals [33]. In another recent review, Marchesi and coworkers (2022) [15] pointed out the potential antinociceptive role of B vitamins, NAC, Curcumin, and Boswellia on NP. B vitamins, for example, could give relief in NP conditions by improving nerve function and regeneration and immune response, increasing the homocysteine (HCY) metabolism, activating the NO-cyclic GMP signaling pathway, reducing the oxidative stress, modulating neural excitability and sodium channel activity in injured dorsal root ganglion (DRG) neurons, and suppressing thermal hyperalgesia [15,34].

NAC may alleviate NP by improving the immune response and reducing oxidative stress, inflammatory responses, and the release of pro-inflammatory substances. Moreover, it may have neuroprotective effects, by modulating the activity of various neurotransmitters and receptors involved in pain transmission. NAC also influences excitatory and inhibitory pain signaling in the central nervous system, by interacting with glutamatergic and GABAergic systems and with the activity of ion channels [35]. Moreover, it may reduce NP, improving the activation of mGlu2/3 receptors and inhibiting the matrix metalloproteinases pathway [15]. 

Curcumin may have a specific effect on NP due to its affinity with the transient receptor potential vanilloid 1 (TRPV1), whose role in the genesis of NP has been found by several studies on animal models [20,21,22]. As reviewed by Caillaud et al. [36], curcumin may have a positive effect on NP thanks to its anti-inflammatory, neuroprotective, and pro-myelination properties as well as due to its capacity to reduce oxidative and endoplasmic reticulum (ER) stress. In a preclinical study [37], curcumin has been found to reduce NP by inhibiting CX3CR1 expression by the activation of NF-κβ p65 in spinal cord and DRG.

Boswellia can alleviate NP by reducing immune and inflammatory responses [15]. Moreover, preclinical studies suggested that extract of this plant can affect NP thanks to its hyperglycemic and antioxidant properties [26], as well as by binding to opioid receptors and activating the NO/cGMP/ATP-sensitive-K+ channel pathway [38]. A recent study on a rat model of sciatic nerve injury also showed that Boswellia can increase neurotrophic factor expression, which could, in turn, promote nerve regeneration [39].

Clinical trials testing the effect of nutraceuticals in humans are rare compared to the greater number of pre-clinical studies in this field, and our results appear, therefore, relevant. Only a few studies have in fact tested the effectiveness of the NACVAN^®^ compounds in humans. B vitamins has been found to relieve different symptoms of NP in a non-interventional study on patients (N = 411) with NP of different etiologies [40], and to reduce the perceived intensity of pain in a descriptive case series on patients (N = 310) with diabetic neuropathy [41]. Instead, a recent parallel, double-blind randomized, placebo-controlled clinical trial showed the efficacy of a curcumin supplementation for 2 months in reducing the severity of diabetic sensorimotor polyneuropathy in patients with type 2 diabetes mellitus [42].

Our study can be relevant also because it has tested the effect of a nutraceutical compound acting as an add-on to gabapentinoids in patients suffering from CNP due to SCI or stroke, i.e., central NP conditions particularly difficult to manage. A recent preclinical study also tested the efficacy of an adjunctive nutraceutical treatment to an ineffective dose of gabapentinoids, showing that the co-administration resulted in a reduction in the NP intensity (thermal hyperalgesia), and suggesting, therefore, an adjunctive effect of the nutraceuticals [43]. 

In recent years, nutraceuticals have garnered significant interest both among patients and clinicians for various factors including: the involvement of inflammation and neuroinflammation mechanisms in the chronification of pain, the absence of adverse effects (much appreciated in patients with multimorbidity), the relative cost-effectiveness, and the possibility of combining them with common pain-related drugs to reduce their dose [44,45]. The possibility of reducing the dose of classic painkillers (e.g., gabapentinoids, opioids) is mandatory, as it can reduce their classic dose-dependent side effects such as the cognitive confusion, dizziness, drowsiness, and constipation, which limit the patients’ quality of life and undermine their treatment adherence. Reducing the dose of gabapentinoids is particularly important in patients with SCI and stroke undergoing intensive neurorehabilitation, as they are sometimes prescribed together with other drugs with central side effects (e.g., benzodiazepines, antidepressants). Such polypharmacology may affect neurological function in clinically relevant ways, favoring, for example, adverse drug reactions, depression, falls, and gait disturbances [46,47], thus, undermining the outcome of the rehabilitation [8,13]. 

The reported findings must be considered with caution given the small number of the recruited patients and the lack of a control group. To further validate these initial findings, future investigations have to encompass a more substantial pool of participants and involve a placebo-controlled group. The lack of a comparison between the effect of NACVAN^®^ in SCI and stroke patients, due to the small number of the stroke patients recruited, represent another study limitation. SCI and stroke are the main causes of CNP and patients with these conditions are often grouped together; however, we cannot exclude a different effect of NACVAN^®^ in the two subgroups of patients [48,49,50]. The results of the present study suggest the need for further pharmacology studies that make us better understand the pharmacodynamic interaction between gabapentinoids and NACVAN. In particular, it may be useful to understand whether the nutraceutical can potentiate the effect of gabapentinoids or exert a positive summatory effect on pain perception.

Even taking into consideration the study limitations, our preliminary observations suggest that NACVAN^®^ could have a beneficial adjuvant effect on pain intensity and on different dimensions of NP, with no adverse effect, among patients with CNP following SCI or stroke. These findings support continued research on the role of NACVAN^®^ as an adjuvant to gabapentinoids in this population. Future analysis on a larger sample, compared with a placebo condition, could confirm these preliminary results that may be of particular interest given the recognized difficulties in the treatment of CNP conditions.

## Figures and Tables

**Figure 1 healthcare-11-02563-f001:**
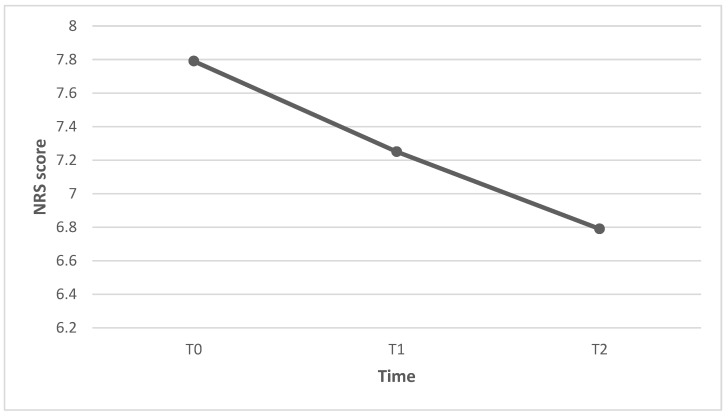
Reduction in pain intensity scores from T0 to T2.

**Figure 2 healthcare-11-02563-f002:**
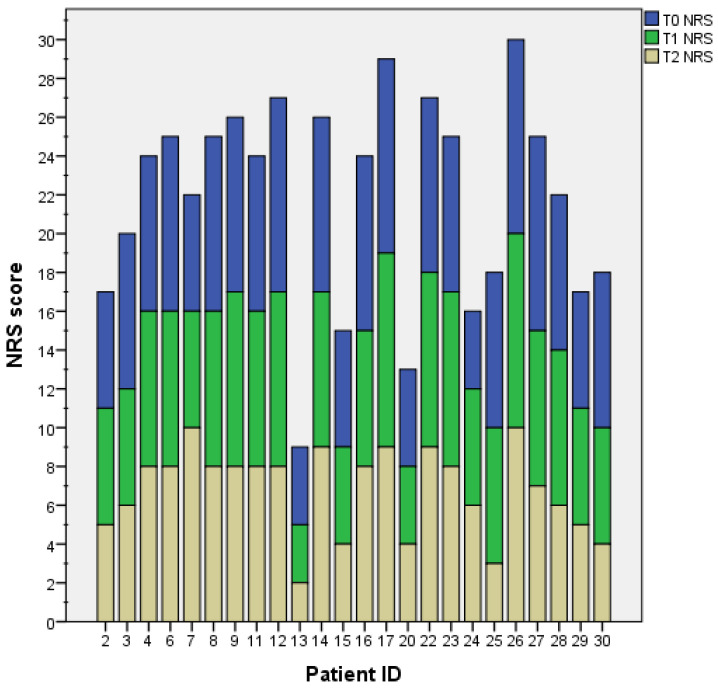
The histogram shows the pain intensity scores measured through the NRS for each patient at each time-point. The two stroke patients are identified with IDs 2 and 4.

**Figure 3 healthcare-11-02563-f003:**
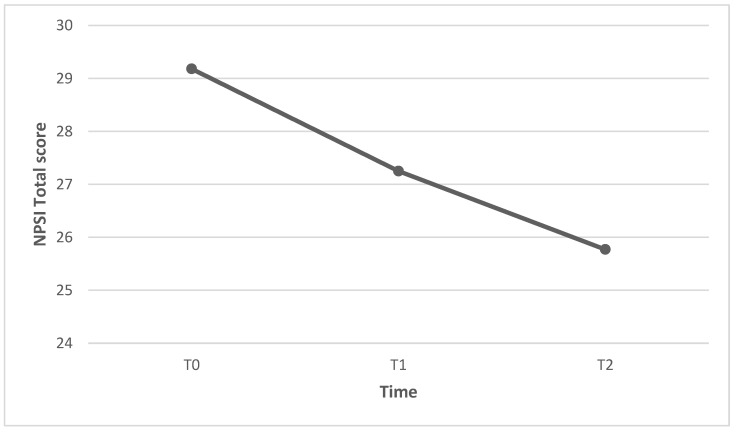
Reduction in NPSI total scores from T0 to T2.

**Figure 4 healthcare-11-02563-f004:**
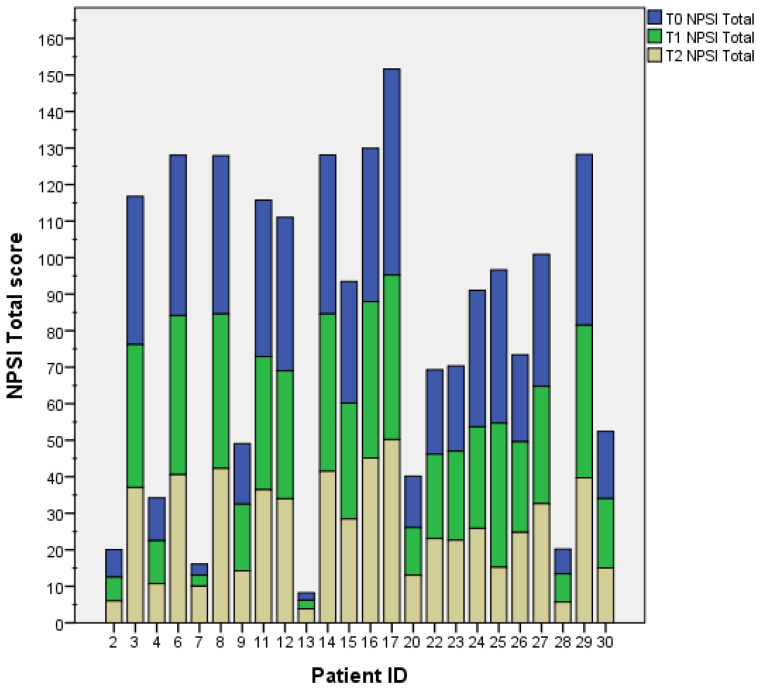
The histogram shows the NPSI total scores for each patient at each time-point. The two stroke patients are identified with IDs 2 and 4.

**Figure 5 healthcare-11-02563-f005:**
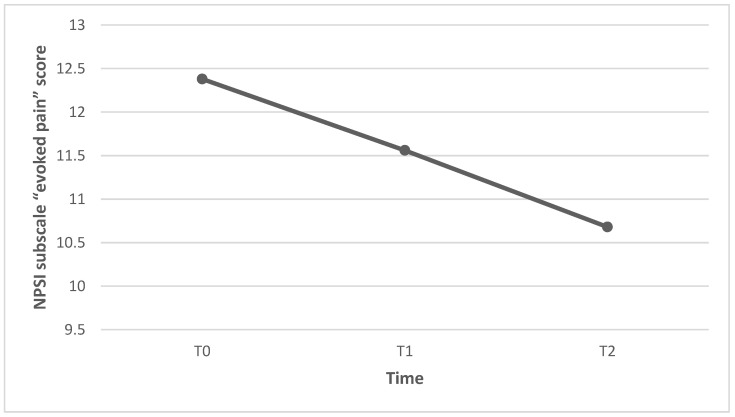
Reduction in NPSI subscale “evoked pain” scores from T0 to T2.

**Figure 6 healthcare-11-02563-f006:**
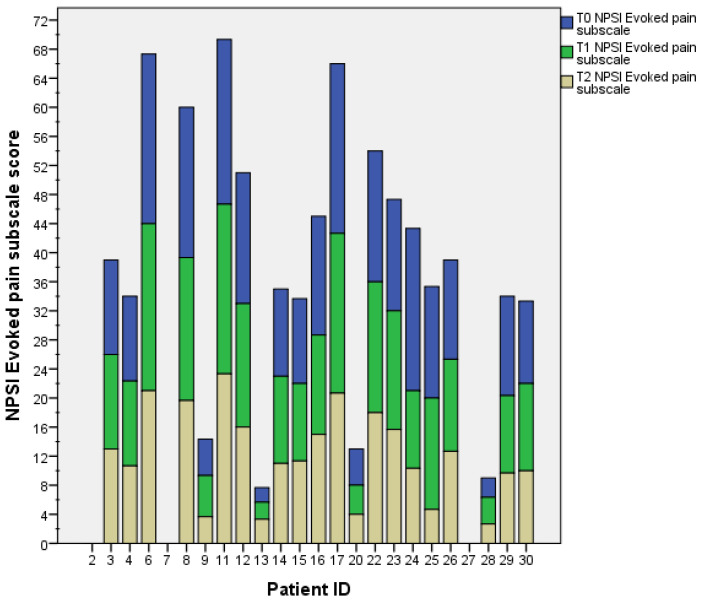
The histogram shows the NPSI evoked pain subscale scores for each patient at each time-point. The two stroke patients are identified with IDs 2 and 4.

**Figure 7 healthcare-11-02563-f007:**
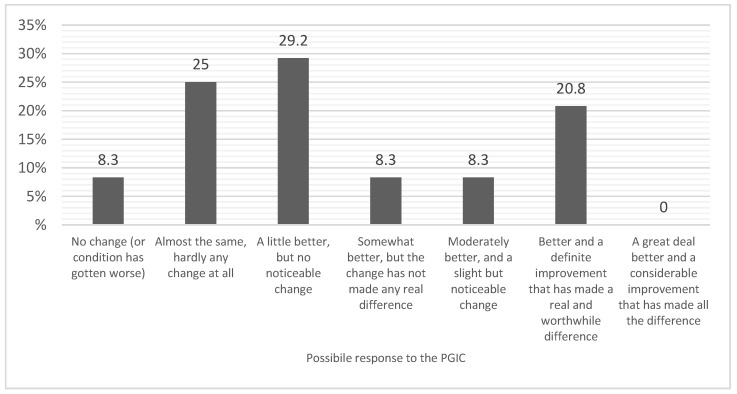
Patients’ global impression of change after 6 weeks of NACVAN intake. The figure shows, for each possible response to the PGIC, the percentage of patients who gave that particular response.

**Table 1 healthcare-11-02563-t001:** Baseline socio-demographic and clinical characteristics of the 24 adult patients with neuropathic pain who received the NACVAN^®^ for 6 weeks.

Characteristic	Value
Age, years	63.67 ± 15.47
Sex, male	14 (58.3)
Education	
*Primary to 3 years professional school*	16 (66.6)
*High school*	6 (25)
*University degree*	1 (4.2)
*Postgraduate*	1 (4.2)
Occupation	
*Employed*	11 (45.8)
*Unemployed*	1 (4.2)
*Housewife*	2 (8.3)
*Retired*	10 (41.7)
SCI	22 (91.7)
Stroke	2 (8.3)
Onset of pain	
*Coincident with the spinal injury or stroke*	24 (100)
*Occurred months after the spinal injury or stroke*	0 (0)
Pain location *	
*Pain at the injury site*	12 (52.2)
*Pain at the injury site with irradiation to a single hemibody*	4 (17.4)
*Pain at the injury site with bilateral sublesional irradiation*	7 (30.4)
Pain frequency	
*Intermittent*	8 (33.3)
*Continuous*	16 (66.7)
SCI-related surgeries *	
*None*	6 (26.1)
*One*	9 (39.1)
*More than one*	8 (34.8)
Pharmacotherapy	
*Pregabalin 25 mg one daily*	1 (4.2)
*Pregabalin 50 mg 3 times daily*	1 (4.2)
*Pregabalin 75 mg once daily*	4 (16.7)
*Pregabalin 75 mg twice daily*	6 (25)
*Pregabalin 75 mg 3 times daily*	2 (8.3)
*Pregabalin 100 mg 3 times daily*	3 (12.5)
*Pregabalin 100 mg 4 times daily*	1 (4.2)
*Pregabalin 150 mg once daily*	1 (4.2)
*Pregabalin 150 mg twice daily*	3 (12.5)
*Gabapentin 100 mg twice daily*	2 (8.3)
Perceived efficacy of pharmacotherapy	5.13 (2.14)

Data presented as mean ± SD or n (%) patient prevalence. * 1 missing. The two stroke patients reported pain in the body region coinciding with the brain injury site; in particular, one stroke patient reported pain in the lower hemibody, the other in the upper and lower hemibody.

**Table 2 healthcare-11-02563-t002:** Descriptive data for outcome variables (N = 24).

Variables	Mean (SD)
T0 NRS	7.79 (1.84)
T1 NRS	7.25 (1.80)
T2 NRS	6.79 (2.25)
T0 NPSI	
*Total intensity score*	29.18 (16.03)
*Burning (superficial) spontaneous pain subscale*	3.46 (3.61)
*Pressing (deep) spontaneous pain subscale*	6.13 (4.97)
*Paroxysmal pain subscale*	7.15 (5.94)
*Evoked pain subscale*	12.38 (7.76)
*Paraesthesia/dysesthesia subscale*	7.33 (4.62)
T1 NPSI	
*Total intensity score*	27.25 (14.18)
*Burning (superficial) spontaneous pain subscale*	3.38 (3.50)
*Pressing (deep) spontaneous pain subscale*	5.81 (4.59)
*Paroxysmal pain subscale*	6.44 (5.24)
*Evoked pain subscale*	11.56 (7.23)
*Paraesthesia/dysesthesia subscale*	(7.40 (4.32)
T2 NPSI	
*Total intensity score*	25.77 (14.08)
*Burning (superficial) spontaneous pain subscale*	3.17 (3.40)
*Pressing (deep) spontaneous pain subscale*	5.29 (4.25)
*Paroxysmal pain subscale*	6.56 (5.07)
*Evoked pain subscale*	10.68 (7.15)
*Paraesthesia/dysesthesia subscale*	6.63 (4.33)
T0 SF-12	
*Physical component summary*	33.84 (8.17)
*Mental component summary*	47.73 (15.22)
T1 SF-12	
*Physical component summary*	34.20 (8.21)
*Mental component summary*	48.36 (12.60)
T2 SF-12	
*Physical component summary*	33.89 (7.72)
*Mental component summary*	49.45 (13.31)

## Data Availability

The data presented in this study are available on reasonable request from the corresponding author. The data are not publicly available as the trial is still ongoing.

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
