# Peer review of "A Nutritional Supplement as Adjuvant of Gabapentinoids for Adults with Neuropathic Pain following Spinal Cord Injury and Stroke: Preliminary Results"

_healthcare, 2023, doi:10.3390/healthcare11182563_

Round 1

Reviewer 1 Report

The manuscript offers valuable insights into the potential role of NACVAN® as a complementary therapeutic option for central neuropathic pain (CNP). The study's robust methodology is evident in its inclusion of a precisely defined experimental group comprising 29 adult inpatients affected by CNP resulting from spinal cord injury (SCI) or stroke. The evaluation of treatment outcomes at multiple time points (baseline, 3 weeks, and 6 weeks) permits a thorough assessment of how the intervention's effects evolve over time. The implementation of both repeated measures analysis of variance and the Wilcoxon matched-pairs test imparts rigor to the analysis and reinforces the credibility of the findings.

The study's outcomes are promising. Noteworthy is the initial discovery of substantial reductions in pain intensity, neuropathic symptoms, and evoked pain among the participants. However, it's imperative to acknowledge the study's relatively modest sample size. To further validate these initial findings, it is advisable that future investigations encompass a more substantial pool of participants. Furthermore, exploring the potential for research involving a placebo-controlled group could elevate the level of validation for the observed effects.

In summary, the manuscript provides a noteworthy perspective on the potential of NACVAN® as a supportive treatment for CNP. The strong methodology, promising initial results, and the absence of adverse events collectively position this study as a substantial contribution to the field. Therefore, I support the manuscript for publication, and I am convinced that it will hold interest for clinicians, and researchers, specializing in pain management.

Reviewer 2 Report

Thank you for submitting this novel approach to the treatment of neuropathic pain. Overall the paper had some merit - even though you have presents a non-randomised study without the results from the control group.

However the results as you have presented them require clarification. It was not clear what the results were as regards the three stroke patients. I am not certain that the results as you have presented them warrent your conclusions.

I have made some minor edits to your Enlish language. Some of your sentences were too long and therefore unclear. I have marked in your manuscript the changes I have suggested.

Reviewer 3 Report

Effectiveness of additional NACVAN treatment was shown in this article.

In the discussion part, plausible functions of the constituents of NACVAN are shown. But the discussion of this part was not shown in this article. Please plausible function of each constituent of NACVAN in the nutritional supplemental treatment. Please male the table which indicates the plausible functions of each constituent.

Based on the list, please the flowcharts which suggests the plausible mechanism in the treatment. Please explain the function to abate the neuropathic pain in the treatment of additional NACVAN.  

Please spell out the term of NSPI

Almost good. 

Reviewer 4 Report

Dear author,

Thanks for the submission. This article provided the pre-liminary observations suggesting that NACVAN could have a beneficial adjuvant effect on pain intensity and on different dimensions of NP among patients with CNP following SCI or stroke with no adverse effect. However, there are some issues that still need to be addressed.

1: In 2.1 study design part (Line133), mentioned “This study employs a non-randomized, placebo-controlled, single-blind design with two groups: NACVAN® or placebo”, however, there is no placebo group in this research, all the 24 patients received NACVAN, so please change the wording and revise accordingly.

2: In 2.3 Intervention part (Line 165), only mentioned the 6 weeks treatment duration and there is another time point which is 3 weeks, please revise accordingly.

3: How did you acknowledge the patients about what they might be having before the treatment?

4: For Figure1, Figure 2, Figure 3, should plot the histogram that presents individual data (each patient's data) to make it more scientific sound.  

In general, the most problematic part is the lack of the control group and the sample size is really low as you already mentioned in the discussion part. 

Round 2

Reviewer 2 Report

Thank you for resubmitting your paper. The changes you have made make this a much more readable paper.

There are a few more minor edits I have made (see attached pdf).

Greatly improved although I have made several more corrections and comments within the pdf.

Author Response

We have accepted all your further language revisions. Thank you again for your precious comments and English revisions, which helped us to improve our manuscript.

Reviewer 3 Report

The manuscript was well revised.

Good.

Author Response

Thank you again for your precious comments, which helped us to improve our manuscript.

Reviewer 4 Report

Dear author,

Thanks for the revision. I have no further questions about this manuscript and hopefully you can add more sample numbers as well as the placebo comparision soon.

Author Response

(The authors gave the same response as above.)
